# Determinants of School dropouts among adolescents: Evidence from a longitudinal study in India

**Pradeep Kumar**[1]*, **Sangram Kishor Patel**[2], **Solomon Debbarma**[3], **Niranjan Saggurti**[2]

**1** Research & Data Analysis, Population Council, India Habitat Centre, New Delhi, India, **2** Population Council, India Habitat Centre, New Delhi, India, **3** Magic Bus India Foundation, Hill Side Tuikhuahtlang, Aizawl, Mizoram

* pradeepiips@yahoo.com

**Data Availability Statement:** Data were collected as part of Population Council's UDAYA study which is publicly available on the site of Harvard Dataverse (DOI: 10.7910/DVN/RRXQNT).

## Abstract

### Introduction

India has the largest adolescent population in the world. However, many unprivileged Indian adolescents are still unable to complete schooling. Hence, there is a need to understand the reasons for school dropout among this population. The present study is an attempt to understand the determinants of school dropout among adolescents and identify the factors and reasons that contribute to it.

### Material and methods

Longitudinal survey data- Understanding Adults and Young Adolescents (UDAYA) for Bihar and Uttar Pradesh has been used to identify the determinants of school dropout among adolescents aged 10–19. The first wave of the survey was conducted in 2015–2016, and the follow-up survey in 2018–2019. Descriptive statistics along with bivariate and multivariate analysis was used to observe school dropout rates and factors associated with it among adolescents.

### Results

Results show that the school dropout rate was highest among married girls aged 15–19 years (84%), followed by unmarried girls (46%), and boys (38%) of the same age group. The odds of school dropout among adolescents decreased with an increase in household wealth status. School dropout was significantly less likely among adolescents whose mothers were educated as compared to mothers who had no education. Younger boys [AOR: 6.67; CI: 4.83–9.23] and girls [AOR: 2.56; CI: 1.79–3.84] who engaged in paid work were 6.67 times and 2.56 times more likely to drop out of school than those who were not. The likelihood of school dropout was 3.14 times more likely among younger boys [AOR: 3.14; CI: 2.26–4.35], and it was 89% more likely among older boys [AOR: 1.89; CI: 1.55–2.30] who consumed any substances as compared to those who did not consume any substances. Both younger [AOR: 2.05; CI: 1.37–3.05] and older girls [AOR: 1.30; CI: 1.05–1.62]

**Funding:** The authors received no specific funding for this work.

**Competing interests:** The authors have declared that no competing interests exist.

who acknowledged at least one form of discriminatory practice by parents were more likely to drop out of school than their counterparts. Lack of interest in studies/education not necessary (43%) was the predominant reason among younger boys for school dropout, followed by family reasons (23%) and paid work (21%).

## Conclusions

Dropout was prevalent among lower social and economic strata. Mother's education, parental interaction, participation in sports and having role models reduce school dropout. Conversely, factors such as being engaged in paid work, substance abuse among boys, and gender discriminatory practices towards girls, are risk factors for dropout among adolescents. Lack of interest in studies and familial reasons also increase dropout. There is a need to improve the socio-economic status, delay the marital age of girls, and enhance the government incentives for education, give rightful work to girls after schooling, and provide awareness.

## Introduction

Education is one of the primary determining factors of development for any country [1, 2]. It plays a significant role in enriching people's understanding of themselves and the world. Also, education plays a crucial role in securing economic and social progress and improving income distribution [1]. No country in the world can achieve sustainable economic development without substantial investment in human capital [2]. So, considering the need and importance of the education, targets was set at the global level; in Goal 4 of the Sustainable Development Goals (SDG) framework, which talks about quality of education, and one of the targets of this goal is to ensure that all the girls and boys complete free, equitable, and quality primary and secondary education [3]. Therefore, it is essential to understand how this goal can be achieved and what progress has already been made in this regard. So, pertaining to this; the scenario at the national level as per National Education Policy (NEP) report indicates that the gross enrolment ratio (GER) for grades 6–8 was 91%, while for grades 9–10 and 11–12, it was 79% and 57%, respectively [4]. Clearly, efforts to bring children within the formal education system through primary schooling have been successful. However, the increasing dropout rates among Indian children, especially after 8th grade, has put the long-term benefits of such gross enrolment into question [5].

A longitudinal study in the US has shown that adolescent employment and school dropouts are strongly associated after adjusting for the individual- and labor-market-level factors [6]. Previous literature has also demonstrated that intensively employed students tend to be less academically successful, less engaged in school, and more likely to drop out [7, 8]. Moreover, research in north Karnataka revealed that economic factors (household poverty; girls' work-related migration) were associated with school dropout among adolescent girls [9]. Another author also substantiates financial obstacles as one of the reasons behind dropout [10].

Poor learning environment and bullying/harassment at school was found associated with an increased odds of school dropout among adolescent girls [9]. While, others factors like distance to school, lack of basic facilities, poor quality of education, inadequate school environment and building, overloaded classrooms, improper languages of teaching, carelessness of teachers, and security problems in girls' schools are major causes of student dropout in

different countries [10]. A cross-sectional community-based study in Raipur, Chhattisgarh, found 11% scholastic dropouts among adolescents [11]. While, poor academic performance is another determining factor [11].

Social norms and practices (child marriage; the value of girls' education) [9], parents' unwillingness [10], socioeconomic status, mother's education, family violence [11] and household's income have significant association with school dropouts [12]. In one prospective study, it was found that social relations were strongly related to the non-completion of secondary education. For example, 18-year-old girls who found family conflicts difficult to handle had a 2.6-fold increased risk of not completing secondary education. Moreover, young people from low-income families were almost three times more likely to not complete secondary education than those from high-income families [13].

Earlier literature has established a link between adolescents engaging in non-academic risky behaviors (e.g. delinquency, drug, alcohol, or cigarette use; sexual involvement, and unintended pregnancies) [14–17], substance abuse [11] and subsequently dropping out of high school [15]. A panel data analysis shows that children whose parents did not participate in Parent-teacher Association (PTA) meetings, discuss academic progress with school teachers, and supervise their children's homework in the first round had a higher risk of dropout in their adolescence (round II) [5]. Poor relations with teachers and classmates at age 18 explained a substantial part of the association between income and dropouts among both girls and boys [13]. A longitudinal study found that students' academic and behavioral engagement and achievement in 10th grade were associated with a decreased likelihood of dropping out of school in 12th grade [18].

Dropout can lead to several consequences as mentioned in the various studies. One of the studies mentioned that dropout from school is an issue that affects not only students who make this decision but also affects their family, the community, and society as a whole [19]. Dropping out of school also leads to under-employment and a lower quality of life for young people [15, 20]. Globally, a large number of students drop out of school every year [21, 22]. While, a significant number of them are found living in poverty or receiving public assistance, imprisoned, unhealthy, divorced, or single parents of children who are likely to repeat the cycle themselves [21, 23, 24]. Dropouts are also at a greater risk of experiencing mental health problems [25] and delinquency [26]. However, it is not clear that risky behavior negatively affects educational achievement and increases the risk of school dropout [27, 28]. One interesting finding from earlier studies reveals that boys who dropped out of school generally worked on family farms, entered the labor market, or undertook vocational training, whereas girls tended to marry [29, 30].

A few decades ago there was a global call to ensure 'education for all' under Millennium Development Goal 2, and now under SDG 4 emphasis is on quality of education; but school dropouts continue to increase in low- and middle-income countries [31]. School dropout is very common in rural India due to various underlying factors. On the other hand, India has the largest adolescent population in the world [32]. This population can benefit the country socially, politically and economically, if they are healthy, safe, educated and skillful. However, many unprivileged Indian adolescents, particularly girls, are still unable to complete schooling. Hence, there is a need to understand the reasons for school dropout among this population. There are a good number of research papers on school dropout in India, but very few focus on the adolescent population. Problems like school dropout can be a major factor in determining adolescents' future perspectives regarding personal and social achievements. The present study is an attempt to understand the determinants of school dropout among adolescents and identify the factors and reasons that contribute to it.

## Data and methods

### Data

This study utilized data from the unique longitudinal survey of adolescents aged 10–19 (Understanding the lives of adolescent and young adults study—hereafter referred to as UDAYA study/survey) in Bihar and Uttar Pradesh. The first wave of the survey was conducted in 2015–2016, and the follow-up survey in 2018–2019. A state-representative sample of unmarried boys and girls aged 10–19 and married girls aged 15–19 was collected in the 2015–16 survey. The study used a multi-stage stratified sampling design to draw sample areas for rural and urban areas separately. In each state, 150 primary sampling units (PSUs)—villages in rural areas and census wards in urban areas—were chosen as the sampling frame, based on the 2011 census list of villages and wards. Households to be interviewed were chosen by systematic sampling in each primary sampling unit (PSU). Each PSU was subjected to a comprehensive mapping and household listing operation (or in selected segments or linked villages as appropriate). The PSU was divided into two nearly equal segments based on the list; one segment was randomly chosen for conducting interviews of females, and the other for interviews of males (married girls were interviewed from both male and female segments). Detailed information about data collection, sampling design of the study has been published elsewhere [33]. The field investigators interviewed 20,594 adolescents using a structured questionnaire; the response rate for the survey was 92 percent, and 1% of selected respondents refused to participate.

In 2018–19, the study re-interviewed those who were successfully interviewed in 2015–16, and who gave consent. The UDAYA study re-interviewed 4,567 boys, and 12,251 girls out of the 20,594 respondents who were eligible. The final follow-up sample consisted of 4,428 boys and 11,864 girls, resulting in an effective follow-up rate of 74% for boys and 81% for girls. The study excluded three percent of respondents who gave inconsistent responses to questions related to age and education during the follow-up survey. The main reasons for loss-to-follow-up were that the participant had migrated (10% for boys and 6% for girls), and the participant or his/her parent or guardian refused (7% for boys and 6% for girls). We note that the characteristics of those who were re-interviewed and those who could not be re-interviewed differed significantly in terms of age, education, place of residence, caste, and religion (see Table 1 in S1 Appendix for attrition bias). The analysis presented in this paper drew on data from the subset of adolescents. The present study considered the sample of adolescents who were enrolled in school at wave 1. The sample size for boys was 3676 and 6178 for girls.

**Outcome variable.** School dropout was the outcome variable of this study. It was defined as a binary variable (yes/no)—whether adolescents dropped out of school between wave-1 and 2. Data pertaining to school dropout was obtained from binary indicators of the school enrolment status collected during both waves of UDAYA. The study included only those adolescents who were enrolled/correspondence in a school during wave 1 [5]. Adolescents who were enrolled in school during wave-1 but not during wave-2 were classified as "yes" (school dropout), while those who were enrolled in both waves were classified as "no" [5].

**Exposure variables.** The explanatory variables included in this study were: place of residence, caste, religion, wealth index, mother's education, engaged in paid work, substance use, state, role model, parental interaction, participation in sports activities, and gender discriminatory practices at home. Place of residence was classified as urban and rural. Caste was categorized as scheduled caste/tribe, other backward class, and others. Religion was grouped into two categories: Hindu and non-Hindu. Household wealth index was constructed based on selected durable goods and amenities with possible scores ranging from 0–57; households were then divided into quintiles, with the first quintile representing households of the poorest wealth

status and the fifth quintile representing households with the wealthiest status [34]. Mother's education was coded as 'no education' and 'educated'. Work status (paid work in last one year) was coded as no and yes. Substance use included consumption of tobacco products, alcohol, and drugs; if the respondent consumed any one of the products, it was coded as "yes", otherwise "no". The survey was conducted in two states—"Uttar Pradesh" and "Bihar". Adolescents reported having a role model (Yes/No). The role models reported were categorized as family members/relatives, teachers, professionals, friends, army/police, sports personalities, friends, actors, politicians and others. Adolescents were considered to have parental interaction (yes/no) if they discussed any of the following topics with their mother or father in the year preceding the interview—school performance, friendship, experience of bullying, physical changes during adolescence, or how pregnancy occurs. Participation in sports activities was coded as 'yes' and 'no'. The respondent was asked—"Do you play any sports or games or engage in physical activities like walking, skipping, running, yoga, etc.?" Respondents were also asked if they experienced any gender discriminatory practices at home where parents favored sons over daughters in any of the following situations—the quantity or quality of food items given, the amount of pocket money given, the type of school in which they were enrolled, and parental aspirations for the respondent's education [34].

**Statistical analysis.** Descriptive statistics were used to observe the school dropout rates among adolescents. Moreover, bivariate analysis was done to find the factors associated with school dropout. A chi-square test was performed to test the significance of the association between outcome variable and predictors of school dropout. Finally, a binary logistic regression analysis was used to observe the relationship between school dropout and other explanatory variables.

The equation for logistic distribution

$$ln\left(\frac{\pi}{1-\pi}\right) = \alpha + \beta_1 X_1 + \beta_2 X_2 + \beta_3 X_3 \ldots \ldots \beta_n X_n$$

Where, $\beta_0, \ldots, \beta_n$, are regression coefficients indicating the relative effect of a particular explanatory variable on the outcome variable. These coefficients change as per the context in the analysis in the study.

**Ethics approval and consent to participate.** The study protocol was approved by the Institutional Review Board of the Population Council. We took several measures to ensure that research ethics were strictly followed. Interviews of boys and girls were undertaken in separate segments of each primary sampling unit to avoid any risk of teasing, harassment and harm to girls' reputation if interviews of boys and girls were conducted in the same geographical segments. Interviews were conducted separately but simultaneously in cases more than one respondent was selected from a household. In order to minimise discomfort during questioning, the scenarios and terminologies described by adolescents were adapted for use in our questionnaire on sensitive topics. Based on our earlier experiences of working with young adolescents, we made the survey questions age-appropriate—for example, we did not ask about sexual and reproductive health matters with young adolescents. Interviewers underwent extensive training in ethical issues, and teams were instructed to apprise community leaders about the study and seek their support for its implementation in the community. Consent was sought from each individual to be interviewed, and for unmarried adolescents aged 10–17, consent was also sought from a parent or guardian. Names were never recorded in the computer form in which data were collected. In order to preserve the confidentiality of the respondent or the parent/guardian, signing the consent form was optional; however, the interviewer was required to sign a statement that she or he had explained the content of the consent form to the respondent or parent. Interviewers were instructed to skip to relatively non-sensitive

sections in case the interview was observed by parents or other family members, call upon a fellow interviewer to conduct parallel discussion sessions with bystander, conduct interviews in locations that offered privacy for the interview and terminate interviews if privacy could not be ensured. Finally, the study team approached NGOs that conduct youth or health-related activities at the district level, help lines that work at national or sub-national levels and public health authorities and referred study participants in need of information or services.

## Results

### Sample distribution of the study population (Table 1)

A higher proportion of adolescents lived in rural areas (81–88%), belonged to Hindu religion (82–96%), and more than half of the adolescents belonged to other backward classes (53–67%). About one-third of the adolescents' mothers were educated (30–40%). A higher percentage of older adolescents engaged in paid work irrespective of their gender. About half of the older adolescents had a role model. A high percentage of adolescents had parental interaction and participated in sports activities.

Fig 1 shows that the overall school dropout rate was highest among married girls aged 15–19 years (84%), followed by unmarried girls (46%), and boys (38%) of the same age group.

Table 2 presents bivariate association of school dropout among adolescents (different age groups and gender) and their background characteristics. Results showed that school dropout was significantly higher among older boys (39%) and girls (49%) who lived in rural areas compared to those who lived in urban areas. Caste has a significant association with adolescents' school dropout. For instance, dropout was more prevalent among adolescents who belonged to SC/ST caste than other castes irrespective of their age and gender. Household wealth has a negative relationship with school dropout among adolescent boys and girls; the dropout was significantly higher among both adolescent boys and girls who belonged to the poorest wealth quintile and it decreases with increase of wealth status of the households. Mother's education also has a significant association with school dropout among adolescents—it was more prevalent among both adolescent (younger and older) boys and girls whose mother had no education. Adolescents who engaged in paid work experienced higher school dropout than those who were not. The most significant difference (paid work and not in paid work) was observed among older boys (59% vs. 33%) and girls aged 15–19 years (42% vs. 21%). Similarly, both younger (41%) and older boys (51%) who consumed any substance had a significantly higher likelihood of school dropout than those who did not. Dropout among younger boys was significantly higher in Bihar (20%), however, it was higher in Uttar Pradesh among married girls (90%). Dropout was lower among adolescents who had any role model irrespective of their age and gender. Parental interaction and participation in sports among unmarried adolescents were significantly associated with school dropout–those who played sports and interacted with parents were less likely to drop out of school.

### Estimates from logistic regression analysis for school dropouts among adolescent boys and girls (Table 3)

The likelihood of school dropout was significantly higher among older girls who lived in rural areas as compared to their urban counterparts [AOR: 1.30; CI: 1.12–1.50]. Moreover, the odds of school dropout were significantly higher among both boys (younger-AOR: 1.77; CI: 1.16–2.69; older-AOR: 1.54; CI: 1.16–2.05) and girls (younger-AOR: 1.78; CI: 1.20–2.64; older-AOR: 1.38; CI: 1.16–1.63) who belonged to a non-Hindu religion as compared to those who belonged to Hindu religion. The likelihood of school dropout among adolescents decreased

**Table 1. Sample distribution of the study population, 2015–16.**

| Background characteristics | Boys aged 10–14 at wave 1 | Boys aged 15–19 at wave 1 | Girls aged 10–14 at wave 1 | Girls aged 15–19 at wave 1 | Married Girls aged 15–19 at wave 1 |
|---|---|---|---|---|---|
| **Place of residence** | | | | | |
| Urban | 15.2 | 17.2 | 17.0 | 18.9 | 12.3 |
| Rural | 84.8 | 82.8 | 83.0 | 81.1 | 87.7 |
| **Caste** | | | | | |
| SC/ST | 25.6 | 26.7 | 24.8 | 20.8 | 19.2 |
| OBC | 57.2 | 52.5 | 56.2 | 54.4 | 66.6 |
| Others | 17.2 | 20.8 | 19.0 | 24.9 | 14.2 |
| **Religion** | | | | | |
| Hindu | 86.1 | 87.1 | 81.7 | 84.7 | 95.6 |
| Non-Hindu | 13.9 | 12.9 | 18.3 | 15.3 | 4.4 |
| **Wealth Index** | | | | | |
| Poorest | 15.0 | 5.8 | 14.9 | 7.1 | 5.2 |
| Poorer | 21.9 | 16.2 | 19.0 | 13.7 | 13.2 |
| Middle | 21.7 | 22.8 | 20.0 | 20.4 | 19.5 |
| Richer | 21.1 | 26.1 | 25.0 | 26.0 | 33.6 |
| Richest | 20.3 | 29.1 | 21.1 | 32.7 | 28.6 |
| **Mother's education** | | | | | |
| No education | 71.1 | 64.9 | 67.5 | 60.0 | 70.5 |
| Educated | 28.9 | 35.1 | 32.6 | 40.0 | 29.5 |
| **Engaged in Paid work** | | | | | |
| No | 91.7 | 79.4 | 93.5 | 85.7 | 93.5 |
| Yes | 8.3 | 20.7 | 6.5 | 14.3 | 6.5 |
| **Substance use** | | | | | |
| No | 96.7 | 86.4 | 99.2 | 98.9 | 99.2 |
| Yes | 3.3 | 13.6 | 0.8 | 1.1 | 0.8 |
| **States** | | | | | |
| Uttar Pradesh | 65.1 | 66.4 | 63.9 | 70.2 | 54.1 |
| Bihar | 34.9 | 33.6 | 36.1 | 29.8 | 45.9 |
| **Role model** | | | | | |
| No | 61.5 | 48.5 | 61.2 | 57.2 | 58.7 |
| Yes | 38.5 | 51.5 | 38.9 | 42.8 | 41.3 |
| **Parental interaction** | | | | | |
| No | 14.6 | 16.2 | 11.0 | 7.5 | - |
| Yes | 85.4 | 83.8 | 89.0 | 92.5 | - |
| **Participated in sports** | | | | | |
| No | 5.4 | 12.7 | 22.1 | 44.8 | 70.9 |
| Yes | 94.6 | 87.3 | 77.9 | 55.2 | 29.1 |
| **Gender discrimination** | | | | | |
| No | 92.2 | 92.6 | 85.1 | 90.6 | - |
| Yes | 7.8 | 7.4 | 14.9 | 9.4 | - |
| **N** | 1619 | 2057 | 1280 | 4213 | 685 |

Note: Wave 1 refers to 2015–16

with an increase in household wealth status. Mother's education plays a significant role in reducing school dropout among adolescent boys and girls and married girls. School dropout was significantly less likely among adolescents whose mothers were educated as compared to

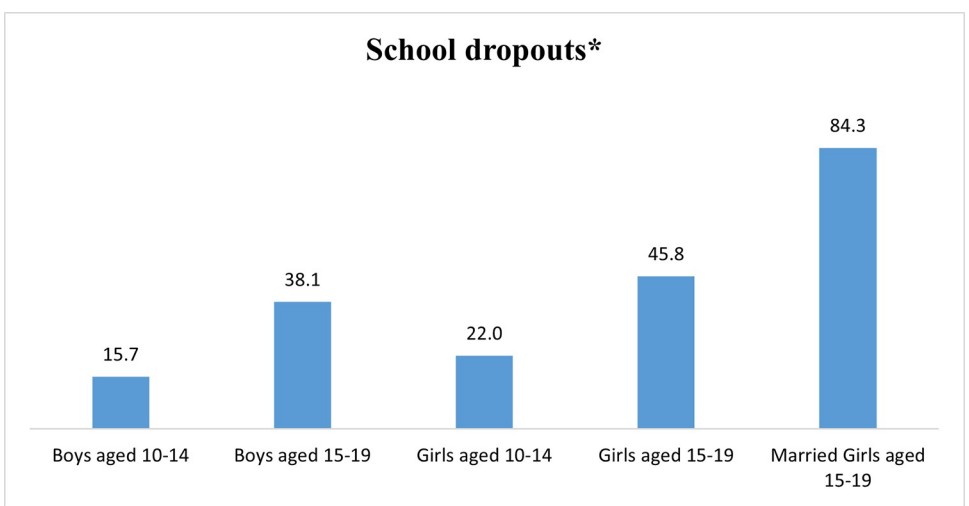

**Fig 1. School dropout among adolescent boys and girls, and married girls, 2018–19.** *overall dropouts: those who were in schooling/correspondence at wave 1 and discontinued at wave 2.

mothers who had no education. Younger boys [AOR: 6.67; CI: 4.83–9.23] and girls [AOR: 2.56; CI: 1.79–3.84] who engaged in paid work were 6.67 times and 2.56 times more likely to drop out of school than those who were not. Similarly, the risk of school dropout was significantly more likely among older boys who engaged in paid work than those who were not engaged in paid work [AOR: 2.86; CI: 2.35–3.49]. The likelihood of school dropout was 3.14 times more likely among younger boys [AOR: 3.14; CI: 2.26–4.35], and it was 89% more likely among older boys [AOR: 1.89; CI: 1.55–2.30] who consumed any substances as compared to those who did not consume any substances. The odds of school dropout were 65% higher among younger boys who belonged to Bihar [AOR: 1.65; CI: 1.21–2.27]. The risk of school dropout was 22% and 13% less likely among older boys and girls, respectively, who had a role model than those who did not have. Moreover, parental interaction and participation in sports activities were significant predictors of dropout among adolescents. Both younger [AOR: 2.05; CI: 1.37–3.05] and older girls [AOR: 1.30; CI: 1.05–1.62] who acknowledged at least one form of discriminatory practice by parents were more likely to drop out of school than their counterparts.

Reasons for school dropouts among adolescent boys and girls, and married girls (Table 4)

Lack of interest in studies/education is not necessary (43%) was the predominant reason among younger boys for school dropout, followed by family reasons (23%) and paid work (21%). Among older boys, paid work (32%) was the primary reason for school dropout, followed by lack of interest in studies/ education not necessary (29%). Among younger girls, family reasons (31%) were the main factor for school dropout, followed by school-related reasons (31%) and lack of interest in studies/ education is not necessary (26%). In contrast, school-related reasons (32%) played a significant role in school dropout among older girls, followed by family-related reasons (26%) and failures (23%). Among married girls, getting married/ engaged (38%) was the major reason for school dropout, followed by failures (25%) and family-related reasons (23%).

Adolescents who lived in rural areas, belonged to SC/ST caste group, belonged to the poorest wealth quintile, and whose mother was not educated, reported more family-related reasons for school dropout compared to their counterparts irrespective of their gender and marital status. Similarly, personal reasons for school dropout were reported more by unmarried

**Table 2. Bivariate association of socio-economic and demographic factors with school dropouts among adolescent boys and girls, 2018–19.**

| Background characteristics | Boys aged 10–14 at wave 1 | Boys aged 15–19 at wave 1 | Girls aged 10–14 at wave 1 | Girls aged 15–19 at wave 1 | Married Girls aged 15–19 at wave 1 |
|---|---|---|---|---|---|
| **Place of residence** | p = 0.313 | **p = 0.027** | **p<0.0001** | **p<0.0001** | p = 0.604 |
| Urban | 13.1 | 33.1 | 12.8 | 33 | 77 |
| Rural | 16.2 | 39.1 | 23.9 | 48.8 | 85.4 |
| **Caste** | **p = 0.004** | **p = 0.003** | **p<0.0001** | **p<0.0001** | p = 0.300 |
| SC/ST | 21.8 | 41.5 | 32.1 | 51.6 | 89.6 |
| OBC | 13.7 | 37.5 | 20 | 47.1 | 84.3 |
| Others | 13.6 | 35.2 | 14.9 | 38.1 | 77.1 |
| **Religion** | **p = 0.004** | **p = 0.006** | **p = 0.039** | **p = 0.008** | p = 0.286 |
| Hindu | 14.8 | 37.2 | 20.1 | 45 | 83.8 |
| Non-Hindu | 21.5 | 44 | 30.4 | 50.6 | 95.6 |
| **Wealth Index** | **p<0.0001** | **p<0.0001** | **p<0.0001** | **p<0.0001** | p = 0.103 |
| Poorest | 19.9 | 53.7 | 40.9 | 63.4 | 83.1 |
| Poorer | 21 | 43.2 | 26.4 | 58.8 | 86.8 |
| Middle | 18.2 | 41.4 | 22.6 | 53 | 85.3 |
| Richer | 11.7 | 37.1 | 16.9 | 44.6 | 87.1 |
| Richest | 8.6 | 30.4 | 10.2 | 33.1 | 79.5 |
| **Mother's education** | **p<0.0001** | **p<0.0001** | **p<0.0001** | **p<0.0001** | **p<0.0001** |
| No education | 18.8 | 41.7 | 27.8 | 52.3 | 87.9 |
| Educated | 8.2 | 31.5 | 10.1 | 36.1 | 75.8 |
| **Engaged in Paid work** | **p<0.0001** | **p<0.0001** | **p<0.0001** | p = 0.367 | p = 0.325 |
| No | 14.5 | 32.6 | 20.6 | 43.5 | 84.2 |
| Yes | 29.4 | 59.2 | 42.1 | 60 | 85.6 |
| **Substance use** | **p<0.0001** | **p<0.0001** | p = 0.099 | **p = 0.025** | p = 0.327 |
| No | 9.9 | 30.4 | 21.3 | 46 | 85 |
| Yes | 41.2 | 51.2 | 32.2 | 43.1 | 70.9 |
| **States** | **p = 0.022** | p = 0.228 | p = 0.427 | p = 0.609 | **p = 0.001** |
| Uttar Pradesh | 13.4 | 37 | 22.8 | 44.8 | 89.8 |
| Bihar | 20.1 | 40.3 | 20.6 | 48.3 | 77.9 |
| **Role model** | p = 0.080 | **p<0.0001** | **p<0.0001** | **p<0.0001** | **p = 0.013** |
| No | 16.1 | 41.6 | 25.4 | 49.7 | 86.8 |
| Yes | 15.1 | 34.7 | 16.8 | 40.7 | 80.8 |
| **Parental interaction** | **p<0.0001** | **p = 0.001** | **p = 0.017** | **p = 0.001** | |
| No | 32.8 | 45.7 | 35.7 | 55.2 | N/A |
| Yes | 12.8 | 36.6 | 20.3 | 45.1 | N/A |
| **Participated in sports** | **p = 0.003** | **p<0.0001** | **p<0.0001** | **p<0.0001** | p = 0.227 |
| No | 26.6 | 47.2 | 37.2 | 52.1 | 84.9 |
| Yes | 15.1 | 36.8 | 17.7 | 40.8 | 82.9 |
| **Gender discrimination** | p = 0.601 | p = 0.297 | **p = 0.007** | **p = 0.005** | |
| No | 11.2 | 29.1 | 17.5 | 37.7 | N/A |
| Yes | 13.8 | 31.8 | 25.4 | 47.2 | N/A |

Note: p-values are based on chi-square test; N/A: Not applicable

adolescents who lived in rural areas, belonged to a lower caste group, and whose mother was not educated (**Table 5**). Moreover, paid work as a reason for school dropout was more reported by boys who lived in rural areas, who belonged to non-Hindu religions, and whose mother was uneducated compared to their counterparts (**Table 6**).

**Table 3. Estimates from binary logistic regression analysis for school dropouts among adolescent boys and girls by background characteristics, 2018–19.**

| Background characteristics | Boys aged 10–14 at wave 1 | Boys aged 15–19 at wave 1 | Girls aged 10–14 at wave 1 | Girls aged 15–19 at wave 1 | Married Girls aged 15–19 at wave 1 |
|---|---|---|---|---|---|
| | AOR (95% CI) | AOR (95% CI) | AOR (95% CI) | AOR (95% CI) | AOR (95% CI) |
| **Place of residence** | | | | | |
| Urban@ | | | | | |
| Rural | 0.63***(0.45–0.87) | 1.0(0.8–1.24) | 1.17(0.82–1.66) | 1.25***(1.08–1.45) | 0.80(0.51–1.24) |
| **Caste** | | | | | |
| SC/ST | 1.23(0.75–2.04) | 1.06(0.78–1.46) | 1.71*(0.99–2.96) | 1.09(0.88–1.35) | 1.20(0.54–2.68) |
| OBC | 0.97(0.63–1.51) | 1.18(0.91–1.54) | 1.02(0.64–1.62) | 1.06(0.9–1.25) | 0.82(0.42–1.57) |
| Others@ | | | | | |
| **Religion** | | | | | |
| Hindu@ | | | | | |
| Non-Hindu | 1.87***(1.27–2.75) | 1.61***(1.21–2.13) | 2.02***(1.35–3.01) | 1.34***(1.13–1.59) | 2.12(0.84–5.35) |
| **Wealth Index** | | | | | |
| Poorest@ | | | | | |
| Poorer | 1.01(0.64–1.58) | 0.64*(0.39–1.05) | 0.59**(0.36–0.96) | 0.76*(0.54–1.05) | 1.24(0.47–3.28) |
| Middle | 0.80(0.51–1.27) | 0.49***(0.31–0.78) | 0.71(0.44–1.15) | 0.63***(0.46–0.86) | 1.06(0.43–2.65) |
| Richer | 0.48***(0.29–0.79) | 0.49***(0.31–0.79) | 0.48***(0.29–0.79) | 0.57***(0.42–0.77) | 0.98(0.4–2.38) |
| Richest | 0.40***(0.22–0.71) | 0.42***(0.26–0.68) | 0.25***(0.13–0.48) | 0.38***(0.27–0.52) | 0.59(0.24–1.47) |
| **Mother's education** | | | | | |
| No education@ | | | | | |
| Educated | 0.49***(0.34–0.72) | 0.80**(0.65–0.99) | 0.43***(0.28–0.66) | 0.63***(0.55–0.73) | 0.57***(0.37–0.86) |
| **Engaged in Paid work** | | | | | |
| No@ | | | | | |
| Yes | 2.08***(1.24–3.48) | 2.51***(1.96–3.2) | 2.03***(1.2–3.42) | 1.53***(1.26–1.86) | 0.86(0.38–1.95) |
| **Substance use** | | | | | |
| No@ | | | | | |
| Yes | 6.41***(3.4–12.11) | 1.95***(1.48–2.56) | 1.63(0.49–5.41) | 0.97(0.51–1.82) | Omitted |
| **States** | | | | | |
| Uttar Pradesh | | | | | |
| Bihar | 1.37**(1.02–1.84) | 0.89(0.72–1.09) | 0.86(0.63–1.19) | 0.99(0.86–1.13) | 0.36***(0.21–0.63) |
| **Role model** | | | | | |
| No | | | | | |
| Yes | 1.03(0.76–1.39) | 0.78**(0.64–0.95) | 0.74*(0.53–1.03) | 0.87**(0.76–0.99) | 0.68*(0.46–1.01) |
| **Parental interaction** | | | | | |
| No | | | | | |
| Yes | 0.50***(0.35–0.71) | 0.78*(0.61–1.01) | 0.58**(0.36–0.92) | 0.74**(0.58–0.93) | N/A |
| **Participated in sports** | | | | | |
| No | | | | | |
| Yes | 0.44***(0.26–0.75) | 0.72**(0.54–0.97) | 0.50***(0.35–0.72) | 0.72***(0.63–0.82) | 0.88(0.58–1.34) |
| **Gender discrimination** | | | | | |
| No | | | | | |
| Yes | 0.88(0.51–1.53) | 1.31(0.91–1.9) | 2.05***(1.37–3.05) | 1.30**(1.05–1.62) | N/A |
| **Constant** | 2.50(0.74–8.51) | 1.88(0.56–6.29) | 1.34(0.54–3.3) | 2.99***(1.76–5.08) | 68.41***(12.77–366.31) |
| **Pseudo R2** | 11% | 7% | 14% | 7% | 7% |

@: reference category

***p<0.0001

**p<0.05

*p<0.10; AOR: adjusted odds ratio; CI: confidence interval; N/A: Not applicable

**Table 4. Reasons for school dropouts among adolescent boys and girls, and married girls, 2018–19.**

| Reasons for school dropouts | Boys aged 10–14 at wave 1 | Boys aged 15–19 at wave 1 | Girls aged 10–14 at wave 1 | Girls aged 15–19 at wave 1 | Married Girls aged 15–19 at wave 1 |
|---|---|---|---|---|---|
| Paid work[†] | 20.9 | 32.2 | 10.2 | 2.2 | 0.6 |
| Family-related reasons[¥] | 22.6 | 20.1 | 30.8 | 26.1 | 23.1 |
| School-related reasons[£] | 15.9 | 16.6 | 31.7 | 31.9 | 12.0 |
| Not interested in studies/education not necessary | 43.3 | 29.2 | 26.3 | 16.1 | 11.6 |
| Illness | 3.1 | 3.0 | 4.2 | 4.8 | 1.5 |
| Failures | 5.3 | 22.0 | 8.6 | 22.6 | 24.6 |
| Got married/engaged | N/A | N/A | @ | 13.2 | 38.1 |
| Others[□] | 8.4 | 4.9 | 6.9 | 6.4 | 17.1 |

[†] included got job and work for payment in cash or kind

[¥] included household work, work on form/family business, care of siblings, and illness or death of a family member

[£] included school too far away, no proper school facilities for boys and girls, transport not available, costs too much, not safe to send girls/boys and poor quality of teaching/education

[€] included illness and not consider education/further education is necessary

[□] included, pregnancy related reason for girls and others; @: frequency less than 25; N/A: not applicable.

**Table 5. Reasons for dropouts among adolescents by socio-demographic and economic characteristics, 2018–19.**

| Background characteristics | Family reasons | | | | | School related reasons | | | | | Not interested in studies | | | | |
|---|---|---|---|---|---|---|---|---|---|---|---|---|---|---|---|
| | Boys aged 10–14 | Boys aged 15–19 | Girls aged 10–14 | Girls aged 15–19 | Married Girls aged 15–19 | Boys aged 10–14 | Boys aged 15–19 | Girls aged 10–14 | Girls aged 15–19 | Married Girls aged 15–19 | Boys aged 10–14 | Boys aged 15–19 | Girls aged 10–14 | Girls aged 15–19 | Married Girls aged 15–19 |
| **Place of residence** | | | | | | | | | | | | | | | |
| Urban | 3.7 | 2.3 | 2.0 | 2.6 | 6.8 | 1.9 | 1.3 | 4.7 | 3.2 | 2.0 | 5.1 | 4.9 | 3.0 | 1.9 | 5.2 |
| Rural | 3.0 | 3.8 | 6.9 | 5.5 | 7.0 | 2.2 | 3.2 | 6.5 | 6.7 | 3.9 | 6.1 | 5.2 | 5.6 | 3.3 | 3.2 |
| **Caste** | | | | | | | | | | | | | | | |
| SC/ST | 5.1 | 4.8 | 9.0 | 8.6 | 9.1 | 2.1 | 5.1 | 8.2 | 8.4 | 3.7 | 9.7 | 7.0 | 7.8 | 3.9 | 1.1 |
| OBC | 2.3 | 3.4 | 5.7 | 4.7 | 7.3 | 2.4 | 2.7 | 7.1 | 6.1 | 3.9 | 4.0 | 4.9 | 4.2 | 3.2 | 3.9 |
| Others | 2.7 | 2.3 | 3.1 | 2.4 | 2.5 | 1.7 | 0.7 | 1.2 | 4.0 | 2.3 | 6.7 | 3.2 | 4.6 | 1.8 | 4.7 |
| **Religion** | | | | | | | | | | | | | | | |
| Hindu | 3.0 | 3.4 | 5.4 | 5.0 | 7.0 | 2.3 | 3.1 | 5.5 | 5.6 | 3.6 | 5.2 | 5.1 | 4.5 | 3.1 | 3.6 |
| Non-Hindu | 3.8 | 4.4 | 9.0 | 4.9 | 6.2 | 1.5 | 1.8 | 9.3 | 8.4 | 5.3 | 10.7 | 5.3 | 8.1 | 2.5 | 1.0 |
| **Wealth Index** | | | | | | | | | | | | | | | |
| Poorest | 4.1 | 6.9 | 12.0 | 15.1 | 13.9 | 1.7 | 7.5 | 14.4 | 11.8 | 2.7 | 7.2 | 5.6 | 11.2 | 9.7 | 5.0 |
| Poorer | 4.1 | 6.2 | 9.6 | 7.0 | 7.2 | 4.0 | 4.3 | 9.2 | 9.7 | 7.5 | 8.2 | 6.4 | 6.5 | 4.7 | 5.2 |
| Middle | 4.2 | 3.8 | 6.4 | 5.8 | 10.5 | 2.8 | 3.2 | 5.7 | 7.8 | 4.5 | 7.6 | 5.2 | 5.6 | 4.1 | 3.6 |
| Richer | 1.9 | 1.6 | 4.4 | 4.3 | 6.7 | 0.9 | 1.8 | 2.1 | 6.6 | 3.6 | 3.7 | 4.6 | 3.8 | 2.2 | 4.6 |
| Richest | 1.3 | 2.8 | 0.3 | 1.9 | 3.5 | 1.3 | 1.9 | 3.0 | 1.8 | 1.5 | 3.1 | 4.6 | 0.8 | 0.9 | 1.1 |
| **Mother's education** | | | | | | | | | | | | | | | |
| No education | 3.7 | 4.5 | 8.5 | 6.7 | 8.8 | 2.7 | 2.8 | 8.7 | 8.0 | 4.0 | 7.0 | 6.3 | 6.5 | 4.4 | 3.1 |
| Educated | 1.7 | 1.6 | 1.0 | 2.4 | 2.7 | 1.0 | 3.0 | 1.0 | 3.2 | 2.8 | 3.4 | 2.9 | 2.3 | 1.0 | 4.4 |

**Table 6. Reasons for dropout among adolescents by socio-demographic and economic characteristics, 2018–19.**

| Background characteristics | Paid work | | Failures | | | Got married/engaged | |
|---|---|---|---|---|---|---|---|
| | Boys aged 10–14 | Boys aged 15–19 | Boys aged 15–19 | Girls aged 15–19 | Married Girls aged 15–19 | Girls aged 15–19 | Married Girls aged 15–19 |
| **Place of residence** | | | | | | | |
| Urban | 2.6 | 1.7 | 35.7 | 11.0 | 5.2 | 10.5 | 45.5 |
| Rural | 2.7 | 4.7 | 19.4 | 9.1 | 6.8 | 13.2 | 36.7 |
| **Caste** | | | | | | | |
| SC/ST | 3.4 | 5.9 | 19.8 | 10.7 | 1.7 | 12.4 | 35.1 |
| OBC | 3.5 | 7.0 | 20.9 | 8.7 | 5.8 | 14.6 | 37.4 |
| Others | 0.1 | 1.7 | 26.7 | 8.6 | 23.7 | 7.2 | 47.4 |
| **Religion** | | | | | | | |
| Hindu | 2.8 | 4.5 | 22.5 | 10.3 | 6.8 | 13.0 | 36.1 |
| Non-Hindu | 3.3 | 13.3 | 14.6 | 4.7 | 2.9 | 12.8 | 71.5 |
| **Wealth Index** | | | | | | | |
| Poorest | 3.2 | 21.0 | 14.5 | 14.8 | 10.2 | 16.0 | 27.8 |
| Poorer | 4.2 | 6.4 | 23.1 | 9.8 | 6.6 | 16.6 | 29.9 |
| Middle | 3.4 | 10.4 | 15.9 | 7.1 | 1.0 | 10.7 | 51.0 |
| Richer | 2.4 | 2.0 | 22.8 | 6.1 | 10.9 | 10.6 | 29.3 |
| Richest | 1.2 | 1.7 | 31.4 | 13.3 | 6.6 | 12.2 | 45.0 |
| **Mother's education** | | | | | | | |
| No education | 3.3 | 7.5 | 21.0 | 9.9 | 5.3 | 13.9 | 38.5 |
| Educated | 1.8 | 2.2 | 21.4 | 7.1 | 10.8 | 9.4 | 35.6 |

## Discussion

This study examines the determinants of school dropout among adolescents in Uttar Pradesh and Bihar, based on data from the longitudinal UDAYA (Understanding the lives of adolescents and young adults) study. School dropout cannot be justified by one single reason; rather, it has several contributing factors. The main finding of this study highlights that dropout was high among married girls, and in rural areas, it was high for both boys and girls. Higher the social (Caste) and economic (Wealth quintile) strata lower the dropout rate and children from other religious background (other than Hindu) were found to have higher dropout rates in the study area. Dropout was also high among those who were engaged in paid work. Mother's education and parental interaction were found to reduce dropout rates and the same is true with the participation in sports activities. The main reasons for dropout are 'not interested' in studies, family reasons, paid work and personal reasons.

At the global level, sustainable development goals have identified girl's education as a priority, but the present study among adolescents found that school dropout rate was higher among married girls, followed by unmarried girls and younger boys. There are several possible reasons for this–an earlier study found that in Bihar, girls are married at an early age (Paul, 2021). This is further qualified with the finding that risk of dropout among girls was associated with marriage [35]. Moreover, it was found that Indian households invest equally in boys and girls at primary school level, but at secondary level of education sons are given priority above girls to study further [36]. Costs of education at secondary level are higher, which may be the factor for girls to discontinue [37].

For many health indicators, the reason for rural-urban differential is mainly due to socio-economic status of the household and parent's education [38, 39]. Similarly, we may attribute the higher dropout of older boys and girls in rural areas as compared to their urban

counterparts to the low socio-economic status and parental education. A majority of families in rural areas are economically poor and may have food insecurity, which results in children engaging in farming and household work, thus leading to dropout [40–44].

Caste had a significant association with adolescent school dropout and it was prevalent more among lower social strata. This result may be substantiated by findings of UNICEF & UNESCO 2014, Prakash, Bhattacharjee, Thalinja, & Isac, 2017, wherein higher dropout rates were seen among adolescent girls of low income families living in rural areas, and belonging to a lower caste [9, 45]. Children from different caste groups do not attend classes together, and that can lead to dropout of lower caste groups [46]. Moreover, children of scheduled caste have intrinsic disadvantages that result in less chance of going to school, even after controlling factors like wealth, parental education and motivation, and school quality, etc. [47].

Dropout was higher among adolescent boys and girls who belonged to the poorest wealth quintile—similar results have been found in other studies [10, 37, 48]. Furthermore, poverty interacts with other social disadvantages and pressures vulnerable children to dropout [49]. Mother's education has a significant impact on school dropout. As found in an earlier study, children of educated parents are likely to continue schooling for longer [49]. While a mother's educational level influences length of the girls schooling, it has also been found that illiterate parents are unable to guide their children and that results in low performance and school dropout [42].

This study found that dropout was higher among those who were engaged in paid work rather than unpaid work. As found by Agarwal, many Indian households engage in different kinds of work from an early age to support their families—girls often work as wage laborers and help their mothers in household work, and girls who engage in work frequently remain absent [50]. Time spent on paid or domestic work may leave children with less time for school and learning—as a result, paid work or domestic work leads to school dropout as found in earlier research [51].

The study found that parental interaction among unmarried adolescents plays a significant role in reducing school dropout. Parent-child interaction can help to encourage schooling and to work hard, especially among low social and economically disadvantaged families who otherwise suffer from lack of motivation and low self-esteem. Another significant finding of this study is that participation in sports activities reduced the school dropout among adolescents. This is consistent with the results of the previous literature [52, 53]. Schools/colleges provide the platform to students for sports activities and this might be the reason for fewer dropouts among adolescents who participated in sports activities. Moreover, the present study revealed that both younger and older girls who acknowledged at least one form of favorable discriminatory practices towards boys by parents had higher chances of school dropout. Previous research also shows that gender discrimination is a major reason for school dropout along with poverty and domestic or household responsibilities [54]. The states of Bihar and Uttar Pradesh have a patriarchal value system and an earlier study shows that socio-cultural issues pertinent to gender imbalance, a patriarchal value system, and educational issues disfavored female students [55].

The present study found that engagement in paid work among adolescent boys was the major reason for school dropout. However, among girls, family-related reasons are predominant. Lack of interest in studies was another reason for dropout among adolescents. These findings are consistent with previous literature wherein multiple household duties for girls, early marriage, and poverty were the main reasons for school dropout [56–59]. Conversely, other studies cited financial difficulties as a reason for dropping out for both girls and boys [56, 58, 60].

The study has a few limitations and strengths. This study is based on two Indian states (Uttar Pradesh and Bihar), limiting the generalizability of the findings. The dropout rate may be an overestimate because of the short interval of the survey. Unmeasured factors may have biased the results. For example, information on the educational attainment of adolescents' fathers and their occupations were not available in the study. The variable—'parental interaction' was constructed based on the discussion of following topics—school performance, friendship, experience of bullying, physical changes during adolescence, or how pregnancy occurs— with either the mother or father in the year preceding the interview. There is no direct question related to parental interaction on education matters or their involvement in school activities. Finally, more research is needed to understand the socioeconomic, familial, and other school-related characteristics of adolescents. Despite these limitations, this study has the strengths of a prospective design, the longitudinal nature of data, and large sample size which allows examination of a detailed picture of school dropout, and the use of multiple covariate adjustments.

## Conclusion

In conclusion, it is found that substance use, engagement in paid work, and gender discrimination in families are the risk factors for school dropout. Conversely, factors such as higher economic status, mother's education, having a role model, parental interaction, and participation in sports activities, are protective factors that reduce dropout among adolescent boys and girls. Girl's schooling is a serious concern and there is a need for immediate action. This study found higher dropout rates in rural areas, specifically among girls. From this finding, one could estimate some of the underlying factors of dropout as follows—in rural areas parents are mostly illiterate and unaware about the importance of education, which results in a lack of parent-child interaction. Further, in rural areas, most households are economically poor and socially backward, so this may lead to early child marriage and pressure to engage in paid work. For boys, substance abuse is a major contributing factor towards dropout. Hence, all of these factors directly or indirectly affect dropout, and this study confirms these factors by citing existing literature.

Lastly, to reduce dropout of girls in particular, it is essential to stop child marriages and give awareness to the parents and improve socio-economic status. This can be achieved by giving rightful work to girls after their education, so that both the children and the parents will be motivated. There is also a need for gender sensitivity. The government should give proper awareness and improve girls' incentives for education and introduce some programs that focus on the return of married girls to school.

## Supporting information

**S1 Appendix. The characteristics at wave 1 of adolescents who were re-interviewed and who were not.**
(DOCX)

## Acknowledgments

The authors are grateful to Sanjay Patnaik for his editorial support on the earlier version of this paper. The authors would also like to acknowledge the contributions of other members of the UDAYA study team at the Population Council.

## Author Contributions

**Conceptualization:** Pradeep Kumar.

**Data curation:** Pradeep Kumar.

**Formal analysis:** Pradeep Kumar.

**Investigation:** Sangram Kishor Patel.

**Methodology:** Pradeep Kumar.

**Resources:** Niranjan Saggurti.

**Software:** Pradeep Kumar.

**Supervision:** Sangram Kishor Patel, Niranjan Saggurti.

**Validation:** Pradeep Kumar, Sangram Kishor Patel, Solomon Debbarma, Niranjan Saggurti.

**Visualization:** Pradeep Kumar, Sangram Kishor Patel, Solomon Debbarma, Niranjan Saggurti.

**Writing – original draft:** Pradeep Kumar, Solomon Debbarma.

**Writing – review & editing:** Pradeep Kumar, Sangram Kishor Patel, Solomon Debbarma, Niranjan Saggurti.

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
