## [Decision Letter · Decision Letter 0]

21 Nov 2022

PONE-D-22-13862Transitional School Dropouts among Adolescents: Evidence from a Longitudinal StudyPLOS ONE

Dear Dr. Kumar,

Thank you for submitting your manuscript to PLOS ONE. After careful consideration, we feel that it has merit but does not fully meet PLOS ONE’s publication criteria as it currently stands. Therefore, we invite you to submit a revised version of the manuscript that addresses the points raised during the review process.

We look forward to receiving your revised manuscript.

Kind regards,

Chandan Kumar, Ph.D.

Academic Editor

PLOS ONE

a) Did participants provide their written or verbal informed consent to participate in this study?

Reviewers' comments:

Reviewer's Responses to Questions

**Comments to the Author**

1. Is the manuscript technically sound, and do the data support the conclusions?

Reviewer #1: Partly

Reviewer #2: Yes

2. Has the statistical analysis been performed appropriately and rigorously? 

Reviewer #1: Yes

Reviewer #2: Yes

3. Have the authors made all data underlying the findings in their manuscript fully available?

Reviewer #1: Yes

Reviewer #2: Yes

4. Is the manuscript presented in an intelligible fashion and written in standard English?

Reviewer #1: No

Reviewer #2: No

5. Review Comments to the Author

Reviewer #1: This paper uses a unique longitudinal data set to explore factors underlying school discontinuation in two states of India. Data are state representative and the two waves of the survey were conducted in 2015-16 and 2018-19. Explanatory factors are a range of socioeconomic characteristics, as well as individual characteristics reported in Wave 1, and discontinuation in the intervening period is assessed from Wave 2 data. Findings suggest that substance use, paid work participation, and gendered socialisation place adolescents at elevated risk of discontinuation, while household economic status, maternal education, as well as individual factors such as having a role model, interaction with parents, participation in sports activities have a protective effect.

While this is an interesting topic, my main concern is that currently, the paper is somewhat superficial, hypothesis are not well articulated, and findings not interpreted in sufficient depth. Evidence on five groups of adolescents (10-14, 15-19, unmarried boys and girls, married girls) is provided, but aside from pointing out that married girls are an outlier, gender and age differences suggested by determinants are hardly discussed, and the discussion disappointingly does not offer hypotheses for similarities/differences. Socioeconomic background factors (rural-urban residence, economic status, religion, caste etc are well known factors influencing schooling. But while confirming the relationship using longitudinal data is no doubt interesting, what is new and exciting about these findings is that parent-child indicators (interaction, socialisation, perhaps even mother’s education as a proxy for education) and even individual behavioural factors (substance use, having a role model, engaging in sports) are key factors influencing school discontinuation, even after confounding background factors are controlled. I would strongly recommend that the paper is recast to highlight the importance of these factors in explaining school dropout, if the same relationship emerges when other concerns described below are taken into consideration. These concerns are:

1. I found the title baffling. What is meant by “transitional” school dropouts? The paper does not explain, and I would recommend a clear title, perhaps just “Determinants of school dropout…”

2. The literature review is somewhat disjointed, it needs to be reorganised so as to synthesise what the leading correlates/determinants are, rather than just describe various articles and their conclusions.

3. The dropout indicator needs to be clearer:

a. In Table 2, the three indicators of drop out shown need to be better explained. Is an older adolescent who has completed Class 10 or Class 12 and has discontinued his/her education considered a dropout, and if so, why? Surely a cutoff of Class 10 (or 12) should not denote dropout. Perhaps this is already done, but if so, it is not described. If not done, authors need to redo their analysis, or at most, justify their use of this broad indicator.

b. Table 2 shows three measures of discontinuation – its not clear to me why overall dropout is so much greater than the other two indicators for older adolescents?

c. How is “dropout” operationalised in the multivariate analyses? Three different indicators are provided in Table 2, some clarity needed. If we assume that the minimum required level of education is Class 10 in order for adolescents to make a successful transition to adulthood, then this, or the more stringent dropout before Class 12, should be used.

4. Findings are interesting, showing that even after place of residence, religion/caste and household economic status are controlled, several reflecting parent-child relations (interaction, gendered socialisation) and individual (substance use, engaging in sports, having a role model), and confirming that both domains are important determinants of school discontinuation. However, some refinement would be helpful among the explanatory variables:

a. Parental interaction and gendered socialisation are described as dichotomous indicators, but each comprises a number of areas of interaction or gendered socialisation. Authors need to be clear – does the indicator reflect at least one of these, or does it refer to interaction on all activities probed, or gendered socialisation on all the situations discussed. Table 1 suggests that parental interaction (>80% report interaction) in particular may well be scaled or at least modified to reflect interaction on all/some items.

b. Maternal education is an important determinant of child outcomes, and it would be good, if a sufficient number of better educated women is available, to show at least three categories of education.

c. Just 1-3% of all groups aside from older boys used substances in the first survey. Does it make sense to include this indicator in the study? And what does substance mean – alcohol? Drugs? Tobacco?

5. Could authors include in the analysis any variable(s) that reflect school related obstacles to continuation, as measured in the first survey?

6. Reasons for dropout are interesting, but would authors like to reconsider the reasons clubbed under various headings. For example, “other” represents a huge chunk (19-31%), but the items shown appear to be school related reasons (no transport, cost, not safe…), why not include these as school related reasons? Likewise, education not considered necessary is quite different from illness, and should be separated.

Overall, this is an interesting paper, with new and exciting findings derived from a unique dataset. It has the potential to contribute to what is known about factors influencing premature school discontinuation in India using a far more relevant set of explanatory indicators than are typically available. However, a clear hypothesis needs to be articulated, and interpretation of findings, including gender and age similarities and differences, needs to be more thoughtful. Author may want to clarify some of the comments noted above.

Reviewer #2: The paper examines drop out decisions in India with a focus on Bihar and Uttar Pradesh, two most educationally backward states of India. To their credit, the authors also use longitudinal data. According to the manuscript, between round 1 and round 2 (or 2018-19 vs 2015–16), the UDAYA study had an effective follow-up rate of 74% for boys and 81% for girls.

So those who dropped out from school b/c of out-migration are mostly absent from the very sample that the authors used to model drop out decisions. Therefore, it is important to discuss the implications of lost to follow up from different characteristics on the drop out from school and the study findings.

Further, authors may highlight the key findings in the conclusion section for better understanding of the manuscript results. N/A need to define in the footnote of the tables.

6. PLOS authors have the option to publish the peer review history of their article (what does this mean?). If published, this will include your full peer review and any attached files.

Reviewer #1: No

Reviewer #2: No

---

## [Author Response · Author response to Decision Letter 0]

1 Feb 2023

Reviewer #1: This paper uses a unique longitudinal data set to explore factors underlying school discontinuation in two states of India. Data are state representative and the two waves of the survey were conducted in 2015-16 and 2018-19. Explanatory factors are a range of socioeconomic characteristics, as well as individual characteristics reported in Wave 1, and discontinuation in the intervening period is assessed from Wave 2 data. Findings suggest that substance use, paid work participation, and gendered socialisation place adolescents at elevated risk of discontinuation, while household economic status, maternal education, as well as individual factors such as having a role model, interaction with parents, participation in sports activities have a protective effect.

While this is an interesting topic, my main concern is that currently, the paper is somewhat superficial, hypothesis are not well articulated, and findings not interpreted in sufficient depth. Evidence on five groups of adolescents (10-14, 15-19, unmarried boys and girls, married girls) is provided, but aside from pointing out that married girls are an outlier, gender and age differences suggested by determinants are hardly discussed, and the discussion disappointingly does not offer hypotheses for similarities/differences. Socioeconomic background factors (rural-urban residence, economic status, religion, caste etc are well known factors influencing schooling. But while confirming the relationship using longitudinal data is no doubt interesting, what is new and exciting about these findings is that parent-child indicators (interaction, socialisation, perhaps even mother’s education as a proxy for education) and even individual behavioural factors (substance use, having a role model, engaging in sports) are key factors influencing school discontinuation, even after confounding background factors are controlled. I would strongly recommend that the paper is recast to highlight the importance of these factors in explaining school dropout, if the same relationship emerges when other concerns described below are taken into consideration. These concerns are:

1. I found the title baffling. What is meant by “transitional” school dropouts? The paper does not explain, and I would recommend a clear title, perhaps just “Determinants of school dropout…”

Response: Thanks for the suggestion. Amendment has been done.

2. The literature review is somewhat disjointed, it needs to be reorganised so as to synthesise what the leading correlates/determinants are, rather than just describe various articles and their conclusions.

Response: Modification has been made as per the suggestion in revised manuscript. 

3. The dropout indicator needs to be clearer:

a. In Table 2, the three indicators of drop out shown need to be better explained. Is an older adolescent who has completed Class 10 or Class 12 and has discontinued his/her education considered a dropout, and if so, why? Surely a cutoff of Class 10 (or 12) should not denote dropout. Perhaps this is already done, but if so, it is not described. If not done, authors need to redo their analysis, or at most, justify their use of this broad indicator.

Response: Here in this study, School dropout was defined as whether adolescents dropped out of school between wave-1 and 2. Adolescents who were enrolled in school during wave-1 but not during wave-2 were classified as school dropout, while those who were enrolled in both waves were classified as not dropouts. We have removed other dropouts (10 or 12) for a better understanding of the reader.

b. Table 2 shows three measures of discontinuation – its not clear to me why overall dropout is so much greater than the other two indicators for older adolescents?

Response: We have removed other two measures of school dropouts in the revised manuscript.

c. How is “dropout” operationalised in the multivariate analyses? Three different indicators are provided in Table 2, some clarity needed. If we assume that the minimum required level of education is Class 10 in order for adolescents to make a successful transition to adulthood, then this, or the more stringent dropout before Class 12, should be used.

Response: Authors are agree with your suggestion, however, if we take class 10 as minimum required level of successful transition then we will lose the sample of 10-14 years adolescent as they are not eligible for 10th standard. Keeping this in mind, authors defined dropouts, who were enrolled in school during wave-1 but not during wave-2.

4. Findings are interesting, showing that even after place of residence, religion/caste and household economic status are controlled, several reflecting parent-child relations (interaction, gendered socialisation) and individual (substance use, engaging in sports, having a role model), and confirming that both domains are important determinants of school discontinuation. However, some refinement would be helpful among the explanatory variables:

a. Parental interaction and gendered socialisation are described as dichotomous indicators, but each comprises a number of areas of interaction or gendered socialisation. Authors need to be clear – does the indicator reflect at least one of these, or does it refer to interaction on all activities probed, or gendered socialisation on all the situations discussed. Table 1 suggests that parental interaction (>80% report interaction) in particular may well be scaled or at least modified to reflect interaction on all/some items.

Response: The parental interaction reflects at least one of these (interaction). The number of interactions on all the activities was very less; therefore, we chose at least one interaction on the items. Similarly, for gendered socialization, we took at least one item for the selection. 

b. Maternal education is an important determinant of child outcomes, and it would be good, if a sufficient number of better educated women is available, to show at least three categories of education.

Response: The sample was not enough to make three categories of mothers’ education with five-age cohort of the adolescent therefore authors made it into two group.

c. Just 1-3% of all groups aside from older boys used substances in the first survey. Does it make sense to include this indicator in the study? And what does substance mean – alcohol? Drugs? Tobacco?

Response: Substance use is important indicator or one of the reason for school dropouts. Therefore, we took it as a predictor and results show that adolescent who consumed substances had higher likelihood of school dropout. Substance use included consumption of tobacco products, alcohol, and drugs. It is mentioned in the variable description as well.

5. Could authors include in the analysis any variable(s) that reflect school related obstacles to continuation, as measured in the first survey?

Response: Authors tried to include all possible available factors in the survey, which affect school dropout/continuation.

6. Reasons for dropout are interesting, but would authors like to reconsider the reasons clubbed under various headings. For example, “other” represents a huge chunk (19-31%), but the items shown appear to be school related reasons (no transport, cost, not safe…), why not include these as school related reasons? Likewise, education not considered necessary is quite different from illness, and should be separated.

Response: Thanks for the suggestion. Amendment has been done in the revised manuscript.

Overall, this is an interesting paper, with new and exciting findings derived from a unique dataset. It has the potential to contribute to what is known about factors influencing premature school discontinuation in India using a far more relevant set of explanatory indicators than are typically available. However, a clear hypothesis needs to be articulated, and interpretation of findings, including gender and age similarities and differences, needs to be more thoughtful. Author may want to clarify some of the comments noted above.

Response: Thanks for the suggestion. This study already articulated the important findings coming from this longitudinal data. However, we again look at the findings through the gender and age differences along with covariates lenses and revised further. The hypothesis is clear to us and stated already in paper that there is differences in school drop outs by gender, age and socio-economic and behavioural characteristics. The findings are also very clearly highlighted those in the manuscript. Add further to it, these findings are also linked to the global call to ensure ‘education for all’ under millennium development goal 2, and now under SDG 4 emphasis is on quality of education. These young population can benefit the country socially, politically and economically, if they are healthy, safe, educated and skilful. However, many unprivileged Indian adolescents, particularly girls, are still unable to complete schooling. Hence, there is a need to understand the reasons for school dropout among this population. There are a good number of research papers on school dropout in India, but very few focuses on adolescent population using longitudinal data. Problems like school dropout can be a major factor in determining adolescents' future perspectives regarding personal and social achievements. The present study is an attempt to understand the determinants of school dropout among adolescents and to identify the factors and reasons that contribute to it.

Reviewer #2: The paper examines drop out decisions in India with a focus on Bihar and Uttar Pradesh, two most educationally backward states of India. To their credit, the authors also use longitudinal data. According to the manuscript, between round 1 and round 2 (or 2018-19 vs 2015–16), the UDAYA study had an effective follow-up rate of 74% for boys and 81% for girls.

So those who dropped out from school b/c of out-migration are mostly absent from the very sample that the authors used to model drop out decisions. Therefore, it is important to discuss the implications of lost to follow up from different characteristics on the drop out from school and the study findings.

Response: In UDAYA longitudinal study, in 2018-19, we interviewed again those who were successfully interviewed in 2015-16, and who consented to be re-interviewed. Of the 20,594 who were eligible for re-interview, we re-interviewed 4,567 boys and 12,251 girls. We excluded respondents (3%) who gave inconsistent response to questions related to age and education at the follow-up survey; therefore, the final follow-up sample comprised 4,428 boys and 11,864 girls, thus resulting in an effective follow-up rate of 74% for boys and 81% for girls. The main reasons for loss-to-follow-up were that the participant had migrated (10% for boys and 6% for girls), and the participant or his/her parent or guardian refused (7% for boys and 6% for girls). We note that the characteristics of those who were re-interviewed and those who could not be re-interviewed differed significantly in terms of age, education, place of residence, caste, and religion (see Appendix Table 1 for attrition bias). The analysis presented in this paper drew on data from the subset adolescents.

Appendix Table 1. The characteristics at wave 1 of adolescents who were re-interviewed and who were not 

Baseline Variable Respondents lost to follow up Respondents interviewed in the follow-up sample Mean difference 

Years of education (mean) 7.33 7.37 0.04 

Completed 8 or more years of education (%) 58.70 58.60 0.10 

Currently in School (%) 57.00 64.80 7.8*** 

Mothers level of education (mean) 2.91 2.51 0.40*** 

Place of residence (%) 45.20 57.50 12.3*** 

Social group (% SC\\ST) 21.60 24.30 2.7*** 

Religion (% Hindu) 73.70 80.00 6.3*** 

HH wealth Score (mean) 22.57 21.51 1.06*** 

Total number of respondents 4302 16292 

*** p<0.01, ** p<0.05, * p<0.1

Further, authors may highlight the key findings in the conclusion section for better understanding of the manuscript results. N/A need to define in the footnote of the tables.

Response: Thanks for the suggestion. Amendment has been done.

---

## [Editor Report · Decision Letter 1]

16 Feb 2023

Determinants of School Dropouts among Adolescents: Evidence from a Longitudinal Study in India

PONE-D-22-13862R1

Dear Dr. Kumar,

We’re pleased to inform you that your manuscript has been judged scientifically suitable for publication and will be formally accepted for publication once it meets all outstanding technical requirements.

Kind regards,

Chandan Kumar, Ph.D.

Academic Editor

PLOS ONE
---

## [Editor Report · Acceptance letter]

20 Feb 2023

PONE-D-22-13862R1 

Determinants of School Dropouts among Adolescents: Evidence from a Longitudinal Study in India 

Dear Dr. Kumar:

I'm pleased to inform you that your manuscript has been deemed suitable for publication in PLOS ONE. Congratulations! Your manuscript is now with our production department. 

Kind regards, 

on behalf of

Dr. Chandan Kumar 

Academic Editor

PLOS ONE